# Influence of Professional Values on Attitudes towards Professional Ethics in Future Physical Therapy Professionals

**DOI:** 10.3390/ijerph192113952

**Published:** 2022-10-27

**Authors:** Elena Marques-Sule, Heta Baxi, Anna Arnal-Gómez, Sara Cortés-Amador, Megha Sheth

**Affiliations:** 1Physiotherapy in Motion, Multispeciality Research Group (PTin MOTION), Department of Physiotherapy, University of Valencia, 46010 Valencia, Spain; 2Self Employed Women’s Association (SEWA), Ahmedabad 380001, India; 3UBIC Research Group, Department of Physiotherapy, University of Valencia, 46010 Valencia, Spain; 4SBB College of Physiotherapy, Gujarat University, Ahmedabad 380006, India

**Keywords:** attitudes, professional values, ethics, physical therapy, future physical therapy professionals, healthcare

## Abstract

This study aimed to analyze the influence of professional values on attitudes towards professional ethics, as well as the influence of sociodemographic variables on attitudes and professional values in future physical therapy professionals. A total of 231 physical therapy students (53% women; mean age 22.30 (SD = 5.13 years; age range 18–49)) participated. Attitudes towards professional ethics (Attitudes Questionnaire towards Professional Ethics in Physical Therapy) and professional values (Axiological Estimation of Professional Values Questionnaire) were analyzed. Linear regressions were conducted to examine: (i) the statistical prediction of attitudes as a dependent variable, with professional values as independent variables; (ii) whether sociodemographic variables had a relationship with attitudes or professional values. Professional values explained 6.5% of the variance of attitudes towards professional ethics (F(1,230) = 16.08, *p* < 0.001)). In regard to sociodemographic characteristics, age explained 3% of the variance of attitudes (F(1,230) = 7.11, *p* < 0.01) and presence of relatives in healthcare explained 1.9% of the variance in professional values (F(1,230) = 4.35, *p* < 0.05)). These results suggest that an increased awareness of professional values is essential to maximizing the attitudes towards professional ethics in future physical therapy professionals in order to improve their future daily clinical practices.

## 1. Introduction

Professional values are the essential elements that build the ethics of a profession and guide the development of the profession. Professional ethics refers to the set of codes of professional conduct, as well as the legitimate purpose of the profession and the way of dealing with ethical conflicts. In Spain, professional ethics in physical therapy are based on the specific ethical principles elaborated by the World Confederation of Physiotherapy [1], of which the Spanish Association of Physical Therapists is a member. Within the framework of professional ethics, professional values are the essential elements on which the standards of action are drawn up and the frameworks for professional ethical action are established [2].

Students come to university with their own personal values, which are the result of their life experience. However, these will not guarantee the development of professional values. The first contact that students have with professional values is during their university education [1,3,4]. Ethics in future physical therapy professionals in the University of Valencia is taught in the second academic year in the compulsory subject ‘Administration, Legislation and Deontology of the Profession’ with 4.5 European Credit Transfer and Accumulation System (ECTS) distributed in a theoretical program (3.5 ECTS) and in a practical program (1 ECTS). The learning strategy is based on participatory master classes, clinical cases, method-based case studies, and online exercises to be solved in groups.

During their educational years and clinical practice, physical therapists encounter a variety of scenarios where a conflict may arise between patient and therapist or between therapist and the other members of the healthcare team, or even amongst the therapists themselves [5]. They negotiate the complexities of participation in multi-professional teams and organizations. In all these relationships and activities, an appropriate understanding of ethics seems to be essential, and there may occur potential situation where conflicts of ethical values may arise. Patients, relatives, and professionals may have quite different views about which interests are the most important ones for each patient. In this regard, the different members of multi-professional teams may value different aspects, and healthcare organizations may appear to value financial targets over quality of care. Professional values may therefore present conflicts with organizational or managerial values [1,6]. These factors may imply the promotion of professional values as a crucial aspect in the education of physical therapists; thus, they can provide health professionals with answers to the ethical dilemmas they face [7].

Despite the importance of professional values in the field of physical therapy, research on professional values is scarce when compared with other health professions [6,7,8,9]. To the best of our knowledge, there is no previous literature that investigates the impact of professional values on attitudes towards professional ethics, as well as the impact of sociodemographic characteristics on attitudes and professional values in physical therapy students. This gap should be addressed since it will help develop new teaching strategies to facilitate the acquisition of professional values among physical therapy students. We hypothesized that professional values could have an impact on attitudes towards professional ethics, and that sociodemographic characteristics of students could also have an impact on attitudes towards professional ethics. 

This study aimed to analyze the influence of professional values on attitudes towards professional ethics, as well as the influence of sociodemographic variables on attitudes and professional values in future physical therapy professionals.

## 2. Materials and Methods

### 2.1. Research Design and Participants

A cross-sectional study was carried out with physical therapy students from the University of Valencia (Spain). This study adheres to the Strengthening the Reporting of Observational Studies in Epidemiology (STROBE) protocol [10]. Participants were voluntary future physical therapy professionals from the physical therapy degree program, aged ≥ 18 years. Participants were excluded if they had prior training in ethics. The study was carried out at the authors’ institution.

### 2.2. Outcome Measures

A total of 650 students from the four courses of the academic degree, both men and women aged ≥ 18 years, were asked to participate by means of snowball sampling, and 231 finally participated (response rate: 35.5%). They were all physical therapy students from the University of Valencia (Spain). After giving informed consent, all the participants provided through an *ad hoc* questionnaire the following data: demographic information; including age and gender; presence of relatives working in healthcare (with a yes/no answer); willingness to choose the degree voluntarily (with a yes/no answer); and working perspectives after finishing the degree (with a yes/no answer). Data were collected from February 2017 to January 2018. Assessments were conducted by a teacher trained in managing the evaluation tools, with more than 10 years of experience in teaching professional ethics. The following outcomes were assessed:

(1)Attitudes towards professional ethics, using the *Attitudes Questionnaire towards Professional Ethics in Physiotherapy* (*AQPEPT*) developed by Aguilar Rodríguez et al. [11]: It is a 33-item self-reported questionnaire in which the questions are divided further into 3 categories: (i) “Importance that future physical therapy professionals give to professional Ethics in Physical therapy degree”, including the following items of the questionnaire: 2, 3, 4, 7, 9, 11, 14, 15, 16, 17, 21, 26, 27, 32; (ii) “Future physical therapy professionals predisposition to train in professional ethics during the degree”, including items 8, 12, 18, 19, 20, 22, 24, 25, 28, 31; and (iii) “Impact of professional ethics training on their professional future”, composed of items 1, 5, 6, 10, 13, 23, 29, 30, 33). Each question was rated on a 5-point Likert scale (1 = Strongly disagree, 5 = Strongly Agree). Internal consistency of this questionnaire has been previously analyzed in physical therapy students (Cronbach’s alpha = 0.898) [11].(2)Professional values, using the Axiological *Estimation of Professional Values Questionnaire* (*AEPVQ*) developed by González-Serna et al. [12], which includes 30 professional values: The importance of each value is registered and rated on an 8-point Likert scale (0 = not important at all, 7 = very important). The higher the score, the greater the importance of the value for the student. The reliability of the internal consistency of the AEPVQ has been previously analyzed, with a Cronbach α = 0.84 for the yes/no question and a Cronbach α = 0.91 for the ranking Likert question.

### 2.3. Ethical Considerations

The study protocol was approved by the Institutional Review Board of the University of the University of Valencia, Spain (H1516821547258). All procedures complied with the Declaration of Helsinki. All enrolled participants were informed of the purpose of the study and written informed consent was taken. 

### 2.4. Statistical Analysis

Descriptive results of continuous data were expressed as mean and standard deviation (SD), while nominal data were described as frequencies and percentages. Correlation analysis was used as the initial step to determine the variables to be included in the regression analysis. Since our data were a mixture of ordinal and nominal data, a Spearman test was used to find the relationship between the total score of AQPEPT and AEPVQ; between the categories of AQPEPT (i.e., importance that future physical therapy professionals give to professional ethics in physical therapy degrees, future physical therapy professionals’ predisposition to train in professional ethics during the degree, and impact of professional ethics training on their professional future) and professional values (AEPVQ); and between attitudes and professional values and sociodemographic variables. Correlation coefficients were interpreted as small, from 0.1 to 0.3, medium, from 0.3 to 0.5, and large, from 0.5 to 1.0 [13]. A linear regression was conducted to examine the statistical prediction of attitudes as a dependent variable, with the professional values as independent variables. Finally, a linear regression was used to test if sociodemographic variables predict the attitudes or professional variables. Statistical analysis was performed using SPSS v. 23.0 (SPSS Inc., Chicago, IL, USA) licensed from the authors’ institution. An external assistant not involved in the study performed the statistical analysis.

## 3. Results

A total of 231 physical therapy students, 122of which being women (53%), agreed to participate and were analyzed. The average age of the participants was 22.30 ± 5.13 years, with age ranging from 18 to 49 years. The demographic characteristics of the participants are depicted in Table 1.

### 3.1. Correlation Analysis

First, a statistically significant correlation was found between AQPEPT total score and AEPVQ total score, resulting in a small positive correlation. Moreover, a significant correlation was found between the AEPVQ total score and the three categories of AQPEPT (i.e., importance that future physical therapy professionals give to professional ethics in the physical therapy degree; future physical therapy professionals predisposition to train in professional ethics during the degree; and impact of professional ethics training on their professional future), with small positive correlation for the three (Table 2).

Second, regarding sociodemographic variables, a significant correlation was found between age and the total score of the AQPEPT questionnaire, resulting in a medium negative correlation. Results also show a significantly small and positive correlation (r = 0.14, *p* = 0.03) between the presence of relatives in healthcare and the total score of the AEPVQ questionnaire. No significant correlations were found amongst the other sociodemographic characteristics and the total score of the AQPEPT questionnaire or the AEPVQ questionnaire (Table 3).

### 3.2. Regression Analysis

Regarding the linear regression analysis, it revealed, in relation to attitudes and professional values, that the model explained 6.5% of the variance and was significant (F(1,230) = 16.08, *p* < 0.001). Regarding attitudes and age, the model explained 3% of the variance and was significant (F(1,230) = 7.11, *p* < 0.01). In relation to professional values and the presence of relatives in healthcare, the results of the regression indicate that the model explained 1.9% of the variance and was significant. (F(1,230) = 4.35, *p* < 0.05) (Table 4). 

## 4. Discussion

To the best of our knowledge, this is the first study that evaluates the influence of professional values on attitudes towards professional ethics, as well as the influence of sociodemographic variables on attitudes and professional values in future physical therapy professionals.

The present study showed that professional values explained 6.5% of the variance in attitudes towards professional ethics reported by future physical therapy professionals and a weak positive correlation between attitudes and professional values. Therefore, our results show that professional values can predict or have an effect on future physical therapy professionals’ attitudes towards ethics. This is an important finding, since attention to ethical considerations in clinical practice seems to be an aspect to be achieved by physical therapy students in order to improve their future practice in clinical settings. Ethics educational training specifically targeting ethical competency could be carried out at the university level in order to help students develop ethics-related aspects. It is also important to take into account that professional values are an integral aspect of professional identity. At the beginning of the history of physical therapy, professional values were a mere combination of their social and personal values rooted in their culture and society. In addition, factors such as family, personal beliefs, environment (University), age, gender, education, and ethnicity have a proven impact on the development of professional values. For this reason, it is necessary for physical therapy students to integrate professional values as ideals of the profession, and thus be able to make consistent and coherent decisions with the ethical standards of the profession [14,15,16,17].

The study performed by Anderson et al. [18] obtained that physical therapy students reported higher total scores for professional values after 33 weeks of clinical practices when compared with the baseline, demonstrating, as reported by McGinnis et al. [7], the possibility of improving professional values after performing clinical practices. Several studies have investigated professional values in other healthcare professions, such as those performed by Poorchangizi et al. [14], which included nursing students who presented higher scores of professional values than practicing nurses. In addition, Nelwati et al. [19] observed that nursing students considered the “caring” dimension as one of the most important professional values. Indeed, all healthcare professionals, including physical therapists, aim at caring for patients; therefore this dimension is important for individuals working as physical therapists in the near future. Moreover, the study performed by Montemurro et al. [20] included medicine students who considered that professional values should be taught more deeply and highlighted the importance of these values when working in clinical settings. Another study carried out by Guenther et al. [21] determined that physical therapists mainly identified as important professional values such as caring, compassion, and integrity. However, although there are some articles that assess professional values in healthcare students or healthcare professionals, there is no literature addressing the specific topic that we explored in our study in physical therapy students, meaning the influence of several ethics-related outcomes in future physical therapy professionals. This implies important aspects to take into consideration, since physical therapy students’ awareness related to professional ethics, as well as the achievement of ethical competence, seem to be important aspects to be achieved by this collective.

The influence of professional values on ethical attitude explains the need to integrate professional values into physical therapy curricula to provide a consistent framework in which all physical therapists follow the same professional values, because today’s society expects to receive physical therapy care that is based on ethical principles and quality. Faculties can help in the development of professional values by acknowledging the difficulties future physical therapy professionals face, while implementing or learning these values and providing solutions that are essential to effectively addressign ethical conflicts [22,23,24].

On the other hand, attitudes towards ethics seem to decrease as future physical therapy professionals grow older. In our results, age was a variable that explained 3% of the variance in attitudes towards professional ethics, as well as a moderate negative correlation between age and attitudes; this means that attitudes towards ethics decrease with age. This is in line with other studies which evaluate future physiotherapy professionals and professional physical therapists. Teachers have to consider these results to try to increase awareness of professional ethics all through the degree. These results are in line with the results of the work by Marques-Sule et al. 2021 [24], in which the younger students gave greater importance to ethics [14]. This is due to the moment in which the ethics course is taught. However, different studies show that professional values change in a positive direction between the beginning and the end of university education. Our results show that professional values can predict or have an effect on future physical therapy professionals’ attitudes towards ethics. For this reason, if learning of professional values were promoted, both in the subject of ethics and in the rest of the physical therapy degree’s subjects, it could be a teaching–learning strategy that favors the development of an ethical attitude in students. In this regard, Venglar and Theall [25] concluded that when students increased their awareness of professional ethics, meaning when they began to know the theoretical and practical aspects of the ethics of the profession, they became aware of the importance and impact that ethics will have on their clinical practice.

The presence of relatives in healthcare explained 1.9% of the variance in professional values, and a weak positive correlation between presence of relatives in healthcare and professional values in future physical therapy professionals was found. The professional values are related with the presence of relatives working in healthcare professions. This is interesting in order to promote professional ethics between future physical therapy professionals. Moreover, it highlights the importance of direct contact with healthcare professionals through the degree. We have not found any scientific evidence in future physical therapy professionals that includes the assessment of the presence of relatives working in healthcare professions. This variable should be taken in account in future studies since it may lead to interesting results.

### Limitations and Strengths

This work has some limitations. First, it should be noted that all the students were from the same institution and from a geographical location in Spain, thus results should be interpreted cautiously. Second, the response rate of our study was low, thus further studies are needed to investigate this issue with larger samples and higher levels of participation of physical therapy students. Third, we did not register the academic year of each participant, although this information would have been interesting to be analyzed. Fourth, our study has a small cohort number, and the age of the participants has to be considered in future research. Fifth, professional values and sociodemographic characteristics may not be the only aspects that can influence attitudes towards ethics; therefore, in future research, other variables could be included. The analysis and scientific evidence on the impact of ethics-related outcomes in future physical therapy professionals is still scarce, so our study offers information regarding this subject. It is important to consider that further studies in the field of physical therapy and professional ethics are needed.

Finally, this study may help to understand the current situation of physical therapy students with regards to professional values and attitudes towards professional ethics. In this regard, this study could be a starting point to explore and develop ethics-related strategies directed to this collective. In addition, it should be taken into account that physical therapy students will be physical therapy professionals who deal with patients in the near future. Thus, assessing ethics-related outcomes and, based on these results, establishing strategies to improve awareness and clinical practice in relation to attitudes and professional values, will help to improve the future daily clinical practice and will provide ethical tools for physical therapists when facing different ethical situations. For this reason, further research on the effectiveness of current ethics teaching would support the implementation of more evidence-based ethics education and training to improve patient care for future physical therapists in clinical settings. Therefore, ethical awareness, ethical tools to deal with ethical problems, and ethical safety in clinical settings could be ensured.

## 5. Conclusions

This study highlights that professional values have an influence on the attitudes towards the professional ethics of future physical therapy professionals. In addition, we observed the importance that future physical therapy professionals give to professional ethics in a physical therapy degree, their predisposition to train in professional ethics during the degree, and the impact that professional ethics training had on their professional future. Moreover, age was inversely related to professional values, while the presence of relatives in healthcare was related to professional values. These results suggest that an increased awareness of professional values is essential to maximizing attitudes towards professional ethics in future physical therapy professionals in order to improve their future daily clinical practice. More research is needed in order to explore how these specific concepts in professional ethics might be linked with practicing physical therapists’ ethical behavior, ethical awareness, and clinical practice, as well as how physical therapists’ actual ethical practice is related to ethical safety in clinical settings and to physical therapy students’ ethical knowledge, awareness, and practice. The assessment of professional values and attitudes towards professional ethics could be considered as an initial measurement of choice in order to explore the appropriate teaching techniques regarding professional ethics in physical therapy curriculums.

## Figures and Tables

**Table 1 ijerph-19-13952-t001:** Sociodemographic variables of the sample.

Sociodemographic Variables	Total (*n* = 231)
Age (years), Mean ± SD	22.30 ± 5.13
Gender, frequency (%)	
Female	122 (53.0%)
Male	109 (47.0%)
Presence of relatives working in healthcare, frequency (%)	
Yes	74 (32.0%)
No	157 (68.0%)
Willingness to choose the degree voluntarily, frequency (%)	
Yes	228 (98.7%)
No	3 (1.3%)
Working perspectives after finishing the degree, frequency (%)	
Yes	228 (98.7%)
No	3 (1.3%)

% = percentage, SD = standard deviation.

**Table 2 ijerph-19-13952-t002:** Correlation between attitudes towards professional ethics and professional values.

Variable	Professional Values (AEPVQ), *r*-Value	*p*-Value
AQPEPT total score	0.26	0.01 *
AQPEPT category 1 score: Importance that future physical therapy professional students give to professional ethics in the physiotherapy degree	0.26	<0.001 **
AQPEPT category 2 score: Future physical therapy professionals students’ predisposition to train in professional ethics during the degree	0.19	0.004 *
AQPEPT category 3 score: Impact of professional ethics training on their professional future	0.26	<0.001 **

AQPEPT = Attitudes Questionnaire towards Professional Ethics in Physiotherapy. * *p* < 0.05, ** *p* < 0.01.

**Table 3 ijerph-19-13952-t003:** Correlation between attitudes and professional values and sociodemographic variables.

Variable	Attitudes (AQPEPT Total Score)		Professional Values (AEPVQ Total Score)	
	*r*-Value	*p*-Value	*r*-Value	*p*-Value
Age	−0.35	0.01 *	0.00	0.99
Gender	0.08	0.22	−0.08	0.23
Presence of relatives working in healthcare	0.12	0.07	0.14	0.03 *
Willingness to choose the degree voluntarily	0.02	0.73	0.02	0.73
Working perspectives after finishing the degree	0.02	0.78	−0.08	0.25

AQPEPT = Attitudes Questionnaire towards Professional Ethics in Physiotherapy. AEPVQ = Axiological Estimation of Professional Values Questionnaire *: *p* < 0.05.

**Table 4 ijerph-19-13952-t004:** Linear regressions performed to examine the prediction of attitudes on professional values, the prediction of age on attitudes, and the prediction the presence of relatives in healthcare predicted for professional values.

Variables	Beta Coefficient	R^2^	F	*t*-Value	*p*-Value
AQPEPT total score—AEPVQ total score	0.256	0.065	16.08	4.01	0.00 *
Age—AQPEPT total score	–0.173	0.030	7.11	–2.66	0.008 *
Presence of relatives in healthcare—AEPVQ total score	0.137	0.019	4.35	2.09	0.04 *

AQPEPT = Attitudes Questionnaire towards Professional Ethics in Physiotherapy. AEPVQ = Axiological Estimation of Professional Values Questionnaire; *: *p* < 0.05.

## Data Availability

Not applicable.

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
