# Peer review of "Influence of Professional Values on Attitudes towards Professional Ethics in Future Physical Therapy Professionals"

_ijerph, 2022, doi:10.3390/ijerph192113952_

Round 1

Reviewer 1 Report

I thank you for the opportunity to review the article entitled: “Influence of professional values on attitudes towards professional ethics in future physiotherapy professionals”. The article is interesting and deals with a current issue, that is, the influence of professional values on attitudes towards professional Ethics, as well as the influence of sociodemographic variables on attitudes and professional values in future physiotherapy professionals.

Although the article is beneficial for the literature, below I propose some suggestions that I hope will help the authors to improve their work.

Introduction. The introduction is clear and the reader feels comfortable entering into the issues discussed. Nevertheless, there is a problem related to the structure of the introduction: this should, in general, start with a general definition, then highlight the gaps in the literature (with recent supporting citations), and finally the research questions should be proposed (see Grant, A. M., & Pollock, T. G. (2011). Publishing in AMJ—Part 3: Setting the hook). What may be lacking at the moment is precisely a discussion of the gaps and how this paper counts on filling them. In fact, at the moment the authors say that the referenced literature is scarce (lines 78-79), yet there are citations from 6 years ago to support it; in the meantime, the world can be said to have completely changed, so I don't know if a 2016 study that says there are no studies on the topic is enough to justify the need for a study like this. It would also be helpful to set the research questions more clearly so that the reader immediately understands what direction the paper wants to take.

Theory and Hypotheses. Although, for that matter, it appears that studies on the topic are scarce, I would appreciate the inclusion of a theory and hypotheses section to put the phenomenon into context. In fact, I think such a section is necessary. Generally, reading the theory should also make it clear what the gap in the literature is and the research demand to support it. Perhaps what is lacking is a clear reporting of the gaps in the literature that you want to fill, and this could improve the readability of the whole text. Perhaps graphically representing hypotheses/research questions could also help create a logical thread in the discourse.

Method. The methodology is ok, and the authors clearly explain the steps they followed. I encourage the authors to detail a bit more about the data collection and analysis process; more specifically, I would appreciate if there was a “Context” paragraph in which the socio-demographic and contextual characteristics precisely of the study are reported. So there is also some information that does not emerge from the current methodology. In particular, I refer to additional details about the sample, e.g., response rate, demographic characteristics, and anything else that may help to understand the methodology adopted.

Results. I applaud the authors for the results, which seem clear and rigorous to me. Kudos.

Discussion, Limitations and Future Research. The discussion is good but it can be improved: the theoretical implications are reported (but it seems more like a repetition of the study results, with even some “numbers” reported, rather than a true discussion of the theoretical implications for the literature), as well as the limitations of the study (but really only a limitation is reported, and only very few references to future research. There is definitely something to be added here). In particular, I believe that this study can provide also important practical implications, and I suggest that more emphasis should be placed on this issue.

I give my praise to the authors for this very interesting study, and I invite the authors to work on the manuscript further to improve its contribution to the literature. I hope that my suggestions could serve this purpose.

Author Response

We would like to thank the reviewer for the comments on our manuscript. For sure the aspects that have been changed due to the comments will help improve the understanding of our research and its impact.

Reviewer 2 Report

Dear Authors,

Thank you for the opportunity to review your paper, describing an important topic. Your are using the concept of professional ethics, but nowhere in the paper has the concept been defined. In order to create clarity around this topic a definition is required. 

Line 21 assessed should be analysed.

Line 27 -30 The conclusion drawn appears not to be related to the findings described. An important finding of this research has been documented as aged explained 3% variance of attitudes. Although, you mentioned that the average age of the participants was 22.30, it is unclear how many participants were younger or older. Drawing a conclusion based on an average age is problematic.

Line 41 Please explain the concept of ethics quality.

Line 43 Which field are you referring to?

Line 45 What do you mean by publishing a code of ethics?

Line 50 There is a plenty  of literature that argues that professional values can not be learned by using theoretical content. Please take this in consideration.

Line 70 Consider replacing methodology with strategy

Line 75 -76 What is the difference between health system and healthcare system?

Line 80 will help to new policies. A word is missing 

Line 92 this sentence is awkward. Consider rephrasing.

Line 98 The data is four years old. Please explain why this is the case.

Line 144 It is unclear if participants were recruited from later years or early years of their degree. This will have an impact on the findings of this study, as you argue in your  paper that students develop their professional values throughout their studies.

The categories in table 2 need clarification, as it is unclear at the moment what has been found.

Line 221 and 222 Not knowing the age of participants makes this argument irrelevant.

The limitation section will need to be expanded including age of data, small cohort number ethics. Line 253 - 257 does not belong in the limitations section. It sound like future recommendations.

The conclusion is very light on and does draw on the findings. This section will need to rewritten.

Author Response

(The authors gave the same response as above.)

Reviewer 3 Report

Dear Authors,

Please find attached my comments about the manuscript.

Best Regards,

Author Response

(The authors gave the same response as above.)

Round 2

Reviewer 1 Report

I have appreciated the new version of your paper. 

Author Response

Dear reviewer, thank you for the time spent reviewing the manuscript. Your contributions have improved the quality of the work.

Best regards  and have a nice day, 

Reviewer 2 Report

Dear authors,

Thank you for responding to my comments and you have addressed them well. In your limitations section you write, Sixth, other factors which  influence ethics and professional values could have been included in this study, although this issue has not been clearly included yet. This sentence does not make sense to me. Please rephrase. 

In your conclusion you write about future research. This section belongs in the discussion, as a conclusion should not contain new information.

Author Response

Dear reviewer, thank you for the time spent reviewing the manuscript. Your contributions have improved the quality of the work.

We have removed the sixth point of the limitations (lines 347-349 ) and corrected the conclusions  (lines 388-390). 

Best regards  and have a nice day, 

Reviewer 3 Report

Dear Authors,

Thanks for incorporating the recommended changes. Now, the article is in a better condition.

Good Luck!

Author Response

Dear Reviewer,

thank you for the time spent reviewing the manuscript. His/Her assessments have significantly improved the quality of the manuscript. 

Best regards and have a nice day